# *Thymus serpyllum* Essential Oil and Its Biological Activity as a Modern Food Preserver

**DOI:** 10.3390/plants10071416

**Published:** 2021-07-11

**Authors:** Lucia Galovičová, Petra Borotová, Veronika Valková, Nenad L. Vukovic, Milena Vukic, Margarita Terentjeva, Jana Štefániková, Hana Ďúranová, Przemysław Łukasz Kowalczewski, Miroslava Kačániová

**Affiliations:** 1Department of Fruit Sciences, Viticulture and Enology, Faculty of Horticulture and Landscape Engineering, Slovak University of Agriculture, Tr. A. Hlinku 2, 94976 Nitra, Slovakia; veronika.valkova@uniag.sk; 2Department of Animal Physiology, Faculty of Biotechnology and Food Sciences, Slovak University of Agriculture, Tr. A. Hlinku 2, 94976 Nitra, Slovakia; petra.borotova@uniag.sk; 3AgroBioTech Research Centre, Slovak University of Agriculture, Tr. A. Hlinku 2, 94976 Nitra, Slovakia; jana.stefanikova@uniag.sk (J.Š.); hana.duranova@uniag.sk (H.Ď.); 4Department of Chemistry, Faculty of Science, University of Kragujevac, 34000 Kragujevac, Serbia; nvchem@yahoo.com (N.L.V.); milena.vukic@pmf.kg.ac.rs (M.V.); 5Faculty of Veterinary Medicine, Institute of Food and Environmental Hygiene, Latvia University of Life Sciences and Technologies, K. Helmaņa iela 8, LV-3004 Jelgava, Latvia; margarita.terentjeva@llu.lv; 6Department of Food Technology of Plant Origin, Poznań University of Life Sciences, 31 Wojska Polskiego St., 60624 Poznań, Poland; przemyslaw.kowalczewski@up.poznan.pl; 7Department of Bioenergy, Food Technology and Microbiology, Institute of Food Technology and Nutrition, University of Rzeszow, Zelwerowicza St. 4, 35601 Rzeszow, Poland

**Keywords:** *Thymus serpyllum*, biofilm, MALDI-TOF MS Biotyper, *Penicillium*, bacillus, stenotrophomonas

## Abstract

The aim of this study was to analyze the chemical composition and biological and antibiofilm activity of the essential oil (EO) of *Thymus serpyllum* with the use of a MALDI-TOF MS Biotyper. The main compounds of the EO were thymol, 18.8%; carvacrol, 17.4%; o-cymene, 15.4%; and geraniol, 10.7%. It was found that free-radical scavenging activity was high. The highest antimicrobial activity was observed against *Pseudomonas aeruginosa*, *Salmonella enteritidis,* and biofilm-forming bacteria. The changes in the biofilm structure after *T. serpyllum* EO application confirmed the inhibitory action and the most pronounced effect was observed on *Bacillus subtilis* biofilm. The antifungal activity of the vapor phase was the most effective against *Penicillium crustosum. T. serpyllum* should be a suitable alternative to synthetic antioxidants as well as antimicrobials. The EO of *T. serpyllum* can be used in the vapor phase in the storage of root vegetables as well as a growth inhibitor of *Penicillium* on bread.

## 1. Introduction

*Thymus serpyllum* or wild thyme of the family *Lamiaceae* is an aromatic flowering plant which contains high amounts of essential oils (EOs) rich in polyphenolic compounds—phenolic acids or flavonoids. EOs are biologically active substances with polyphenolic compounds which exhibit high radical scavenging potential and anti-inflammatory activity [1]. These properties are used in the pharmaceutical industry for pharmaceutical preparations such as antihypertensive, anti-inflammatory, antiproliferative, and anticancer medicines. The origin of *T. serpyllum* is Mediterranean Europe and Africa, and it is typical that the plant vegetation is observed at higher altitudes [2]. *T. serpyllum* extracts possess antibacterial, antimicrobial, antifungal, and insecticidal effects [3]. EOs are known to exhibit this effect on the growth and development of the microorganisms and this is especially important in the prevention of contamination with spoilage and pathogenic microorganisms. *T. serpyllum* EO exhibits a strong inhibitory effect on the growth of Gram-positive and Gram-negative bacteria, and against yeasts [4,5]. *T. serpyllum* EO also has an inhibitory effect against biofilm-forming microorganisms. Inhibition of the biofilm growth has been observed with *Salmonella* Enteritidis [6] and *Enterococcus faecalis* [7].

Biofilm is a bacterial community which forms a structured consortium. Gram-positive, Gram-negative bacteria, and also yeasts strains are able to form biofilm. The biofilm can be attached to surfaces of different structures and functions and it is surrounded by an extracellular polymer matrix produced by the cell. The biofilm occurs in two stages—the planktonic stage and the adherent stage. The formed biofilm is adapted to adverse environmental conditions such as the presence of antimicrobials [8]. The bacteria of the adherent biofilm are more resistant to the antimicrobials used in their elimination than the bacteria of the planktonic stage [9]. The demand for natural alternatives to chemical preservatives, antioxidants, and antimicrobials to ensure food safety and minimize public health hazards is still growing. The negative effects of increased use of synthetic antioxidants on consumer health have been reported [10]. There is an interest in exploring natural alternatives since it is known that the application of chemical agents can result in liver damage as well as an increased incidence of cancers [10].

Microscopic techniques are often used to study morphological and structural changes in bacterial biofilms. The matrix-assisted laser desorption ionization time-of-flight mass spectrometry (MALDI-TOF MS) is used for characterization of different bacterial biofilms and is recognized as a suitable alternative to microscopic techniques [11].

The aims of the present study were to analyze the chemical composition and, antioxidant, antimicrobial, and antibiofilm activity of essential oil of *Thymus serpyllum* from the Slovak region. Evaluation of the inhibitory activity of EO on biofilms by MALDI-TOF MS is a relatively novel technique and our aim was to detect the structural and molecular changes in *Stenotrophomonas maltophilia* and *Bacillus subtilis* biofilm with constructed dendrograms constructed from mass spectra. The ability to detect the effectiveness in the gas phase, in a food model, against biofilm-forming bacteria and *Penicillium* spp. was also tested.

## 2. Results

### 2.1. Chemical Composition of Thymus serpyllum Essential Oil

Gas chromatography/mass spectrometry (GC/MS) and gas chromatography (GC-FID) of *T. serpyllum* EO was used for detection and the major compounds were found to be thymol, 18.8%; carvacrol, 17.4%; *o*-cymene, 15.4%; and geraniol, 10.7% (Table 1).

### 2.2. Antioxidant and Antimicrobial Activity, and Minimum Inhibitory Concentrations (MIC)

The antioxidant activity of *T. serpyllum* measured by the DPPH method was determined at 82.4 ± 0.5% of inhibition which corresponds to 463.53 ± 2.60 TEAC. Weak antimicrobial activity of *T. serpyllum* EO was observed against *Y. enterocolitica*, *S. aureus*, and *C. tropicalis*. Moderate inhibitory activity was found against *B. subtilis*, *E. faecalis*, *C. albicans*, *C. krusei*, and *C. glabrata*. Other microorganisms, including biofilm-forming microorganisms, showed very strong inhibitory activity (Table 2). Using with MIC method, the lowest inhibitory concentrations were found against *S. enteritidis, P. aeruginosa,* and biofilm-forming bacteria. The highest inhibitory concentrations were identified against *Y. enterocolitica*, *S. aureus*, and *C. tropicalis* (Table 2).

### 2.3. Analysis of Biofilm Development and Molecular Differences Using the MALDI-TOF MS Biotyper

Figure 1 shows the spectra of developmental stages of *S. maltophilia* biofilm throughout the experiment.

The essential oil was evaluated according to the results of the mass spectra analysis of *S. maltophilia* with the addition of *T. serpyllum* EO. The spectra of control samples (planktonic bacteria and biofilm untreated with the EO) developed accurately (spectra not provided), thus the control planktonic cells were used as a control for comparison of biofilm structural changes.

Mass spectra after the 3rd day (Figure 1A) of cultivation showed the same peaks which indicates the same protein production by the young biofilms and the control planktonic cells. No changes in bacterial cultures were observed on the protein level.

The difference between mass spectra of biofilms on glass and wooden surfaces and the control sample occurred from the 5th day (Figure 1B–F). There were visible changes in the protein profile of biofilm treated with the EO of *T. serpyllum*. It seems that *T. serpyllum* EO influences the homeostasis of bacterial biofilm formed on the wooden and the glass surface.

The dendrogram was constructed as a visualization of mass spectra for determination of some similarities of biofilm structure regarding the distance of the MSPs. It can be stated from the constructed dendrogram (Figure 2), that the planktonic stage (P) together with control groups and the young biofilms had the shortest distance from the 3rd day when it grew on the wood and the glass (SSM3 and DSM3, respectively). The similarity in protein profile of the control groups was confirmed by the short distances of the MSPs. The young biofilms and control planktonic cells also had short MSP distances which corresponded with the mass spectra. The distance of MSP experimental groups increased gradually over time. Mass spectra prepared on the 12th and 14th day of the experiment had the longest MSP distances which indicated the changes in the bacterial biofilm protein profile of *S. maltophilia.*

Figure 3 shows the spectra of developmental stages of *B. subtilis* biofilm over the entire duration of the experiment.

Mass spectral analysis of *B. subtilis* biofilm showed the similarity of the experimental spectra and the control planktonic spectrum on the 3rd day of the experiment (Figure 3A) which indicates that bacterial biofilm developed equally due to the protein production.

The changes in mass spectra on 5th day were more visible on biofilm formed on wood than on the glass surface (Figure 3B). The changes in the mass spectra, in comparison to control planktonic cells, were observed in biofilm on both surfaces from 7th day (Figure 3C–F).

The changes in protein profile of *B. subtilis* biofilm treated with the EO of *T. serpyllum* were visible two days later than those of *S. maltophilia*. However, the effect of *T. serpyllum* EO on protein production can be confirmed in the biofilm-forming bacteria *B. subtilis* compared to untreated control cells.

The dendrogram constructed according to mass spectra also confirmed the similarity of young biofilms with planktonic cells and control cells. The distance of MSP growth during the experiment progression indicates the differences in protein production caused by influence of *T. serpyllum* EO addition (Figure 4).

### 2.4. Antimicrobial Analysis of Bread in Situ

The vapor phase of the antibacterial activity of *T. serpyllum* EO was confirmed at higher concentrations (Table 3). *S. maltophilia* was inhibited at concentrations of 250 µL/L and 500 µL/L by 83.51% and 44.05%, respectively. *B. subtilis* was inhibited at concentrations of 125–500 µL/L while the highest inhibitory effect of up to 94.47% was achieved at 250 µL/L. Antifungal activity against *P. citrinum* varied between 15.21% and 84.93%. *P. crustosum* was inhibited to a lesser extent than the other fungi tested with the highest inhibition of 54.15% recorded at 500 µL/L. *P. expansum* was the most effectively inhibited by the vapor phase of *T. serpyllum* with an inhibition of 84.93% at 125 µL/L (Table 3).

### 2.5. In Situ Antimicrobial Analysis of Carrots

Inhibition of *S. maltophilia* was recorded on carrots at all concentrations with the highest inhibition rate of 83.77% at 500 µL/L and the lowest of 3.90% at 62.5 µL/L (Table 4). The growth of *B. subtilis* was inhibited on carrots equally at all concentrations. The highest inhibition was 72.46% at 500 µL/L and the lowest was 41.67% at 125 µL/L. The antifungal activity of the vapor phase of *T. serpyllum* EO against *P. citrinum* was 63.27% at a concentration of 125 µL/L. *P. crustosum* was inhibited at all concentrations but the highest inhibition was 53.82% at a concentration of 125 µL/L. The highest inhibition of *P. expansum* was 39.03% at 125 µL/L.

## 3. Discussion

The authors of previously published studies agreed that thymol, cymene, and carvacrol were the most common major components of *T. serpyllum* essential oil. The compounds thymol and carvacrol are known for their potential antimicrobial and antifungal activity [12]. Nikolić et al. [13] identified oxygenated monoterpenes, 54.5%; thymol, 38.5%; monoterpene hydrocarbons, 26.3%; and *p*-cymene, 8.9% as the main compounds of the EO of *T. serpyllum*. The authors state that *T. serpyllum* is very suitable for use in the food industry for food preservation and in the pharmaceutical industry because of its high thymol content. Baj et al. [14] in their study found major compounds such as linalool, 30.9%; thymol, 25.1%; and geraniol, 10.5%. *T. serpyllum* is a suitable alternative to artificial fungicides because of the high content of monoterpene compounds. Kovacevic et al. [15] determined the main compounds of *T. serpyllum* EO to be thymol, 54.17%; γ-terpinene, 22.18%; and p-cymene 16.66%. Thymol (15.21% and 41.8%) was the main compound of *T. serpyllum* EO in the studies of Šojić et al. [16] and Tazabayeva and Sylibaeva [17]. Pruteanu et al. [18] also found that thymol, 36.30%; *o*-cymene, 24.98%; and hydroquinone tert-butyl, 10.25% were the main components of *T. serpyllum* EO. Our results are in line with above-listed authors. *T. serpyllum* is suitable for use in food preservation due to the dominant presence of thymol and carvacrol in its EO. The differences in the content of the main compounds in previous studies could be attributed to different origins of the EO. *T. vulgaris* or *T. zygis* had minimum thymol and carvacrol contents of 40% [19] which was similar to the results of our study. The *Council of Europe and European Medicines Agency Assessment Report* lists concentrations of thyme EO as 37–55% thymol and 0.5–5.5% carvacrol, what is higher in comparison with our study [19,20].

The method for DPPH radical scavenging is variable in terms of solvent, incubation time, ratio of sample, and DPPH solution which makes the proper comparison of the present results with previous studies more difficult. Nikolić et al. [13] determined DPPH scavenging activity of *T. serpyllum* EO with different ratios of the sample and DPPH. With 30 μL of sample and 270 μL of DPPH the results were determined as EC 50 at a value 0.96 μg Trolox/mL; thus the activity of *T. serpyllum* EO was strong. Kulisic et al. [21] detected the percentage of inhibition of *T. serpyllum* EO oil at 82.00 ± 0.07% for DPPH and the sample in ethanol, and antioxidant activity was considered as strong. Hussain et al. [22] determined the antioxidant activity value as IC50 at 34.8 ± 1.9 μg/mL after incubation for 1 h which was evaluated as good activity. All authors concluded that the antioxidant activity of *T. serpyllum* EO is strong that supports our results despite of the variations in method and the differences in the results.

Hussain et al. [22] found a strong antimicrobial effect of *T. serpyllum* against human, plant, and foodborne pathogens using the disc diffusion method. They determined strong inhibitory activity for various microorganisms. The inhibition zones for microorganisms were slightly higher—*S. aureus* (20.2 ± 0.7 mm), *B. subtilis* (27.2 ± 0.9 mm)*, P. aeruginosa* (16.2 ± 0.6 mm)*, and Salmonella* sp. (20.8 ± 0.5 mm) —compared with our study. Goja et al. [23] confirmed the strong antimicrobial activity of *T. serpyllum* with inhibition zones of *S. aureus,* 34 ± 1 mm; *P. aeruginosa,* 7 ± 1 mm; and *C. albicans,* 26 ± 1 mm. *Thymus serpyllum* EO is suitable as a source of natural food additives due to its diverse antimicrobial effect. Ouedrhiri et al. [24] identified antimicrobial activity of *T. serpyllum* with inhibition zones for *S. aureus* of 36.00 ± 1.73 mm, *B. subtilis* of 33.00 ± 2.64 mm, and *P. aeruginosa* of 10.00 ± 1.73 mm. *T. serpyllum* EO showed strong bactericidal activity against a diverse range of microorganisms; it contains increased thymol content, which can be responsible for the antibacterial effect. Ouedrhiri et al. [24] determined the MICs of *T. serpyllum* EO for *S. aureus,* 0.25 μL/mL; *B. subtilis,* 0.125 μL/mL; and *P. aeruginosa,* 4 μL/mL. These values are low and confirm the strong antimicrobial effect of *T. serpyllum*. MIC 50 and MIC 90 were 2 ± 0.2 μL/mL and 4 ± 0.3 μL/mL for *C. albicans* in the study by Nikolić et al. [13]. The determination of a low minimum concentration for *C. albicans* corresponds to our findings on the effect of *T. serpyllum* against the genus *Candida*. Wesołowska et al. [25] determined MICs of 0.78 μL/mL for *S. aureus* and of 1.56 μL/mL for *E. faecalis*. This finding indicates the possible use of *T. serpyllum* EO as an antimicrobial for food preservation and for the prevention of bacterial and fungal infections. Inhibitory activity was reported in the study by Hussain et al. [22] where MICs of 0.160 ± 0.003 μL/mL for *S. aureus*, 1.25 ± 0.034 μL/mL for *P. aeruginosa,* and 0.83 ± 0.02 μL/mL for *Salmonella* spp. were found. It is known that *T. serpyllum* is considered as a suitable alternative to synthetic antimicrobials which confirms our findings.

The changes in MALDI-TOF mass spectra of treated and untreated sample were visible show the differences in protein production. Various research has proposed the abnormalities in protein production related to biofilm formation and degradation after EO addition. Nazzaro et al. [26] described the mechanisms of action of the EOs in damaging cell components and cell mechanisms, and in protein degradation. Szabó et al. [27] considered that EOs affect the quorum sensing in biofilm-forming bacteria, that could lead to changes in the cell at the protein level. Camele et al. [28] described that the quorum sensing mechanism is deteriorated by the influence of EOs and that disrupting the quorum sensing mechanism can help to prevent biofilm formation. Moreover, the differences in protein production of bacteria stressed with EOs was evaluated with MALDI-TOF MS by Božik et al. [29] demonstrating that EOs affect the production of ribosomal proteins and stress-related, membrane-related, and biofilm-related proteins.

The use of MALDI-TOF for detection of degradation of biofilm has been previously less reported. Kırmusaoğlu [30] described various methods for biofilm detection and stated that mass spectrometry is a less common but very suitable method for biofilm research. Stîngu et al. [31] described the accuracy of identification of biofilm-forming bacteria by MALDI-TOF MS compared to 16S rRNA sequencing and confirmed that MALDI-TOF MS can distinguish the differences in the mass spectra of closely related biofilms. Gaudreau et al. [32] analyzed 18 biofilm-producing species by MALDI-TOF MS. They were able to identify 72% of bacteria at species level and 83% at genus levels with high confidence. Aguiar et al. [33] detected specific biofilm proteins with MALDI-TOF MS and suggested the suitability of using MALDI-TOF MS for biofilm-forming yeast detection. Kačániová et al. [34,35] demonstrated the changes in protein profile analysed by MALDI-TOF MS in their previous studies and the inhibitory effects of *Coriandrum sativum* and *Citrus aurantium* EOs on antimicrobial activity and biofilm formation. The inhibitory activity of EOs can support the idea that the changes in protein spectra could be connected with the structural and molecular changes of biofilm formation, growth, and degradation so MALDI-TOF MS can be used for these analyses. These findings correspond with current results. We can also confirmed that the structural changes of biofilm are well detected and MALDI-TOF MS can be used for detection of the changes in the protein profile of biofilm. The mechanisms of action of *T. serpyllum* EO at the protein level could be the main topic of further research.

The confirmed antibacterial effect of *T. serpyllum* shows the potential for it to be used in the storage of carrots and it could be considered as a natural alternative to chemical inhibition of *Penicillium* growth on bread. The effect in the vapor phase was described in the report by Střelková et al. [36]. Feng et al. [37] tested the antifungal effect and found that the vapor phase treatment is more efficient, faster, and easier to use in comparison to soaking.

The direct contact of Eos on microorganisms has often been studied in the past. Direct contact has demonstrated the antimicrobial effect of many EOs. There are only a few studies on the use of the vapor phase of EOs for the inhibition of microorganisms. *S. maltophilia* was inhibited at concentrations of 250 µL/L and 500 µL/L. *B. subtilis* was inhibited at concentrations of 125–500 µL/L. In correspondence with our results, **a** significant antimicrobial effect of thymus EO in the vapor phase was observed against several pathogenic bacteria [38,39,40,41,42]. The antifungal activity against *P. citrinum* varied from 15.21% to 84.93%. *P. crustosum* was inhibited to a lesser extent than the other fungi tested with the highest inhibition of 54.15% at 500 µL/L. *P. expansum* was the most effectively inhibited by the vapor phase of *T. serpyllum* with 84.93% inhibition at 125 µL/L. The different dose of essential oils that are effective against different microscopic fungi were found in several studies [43,44,45,46,47]. These doses varied from 40 to 300 µL/L against *Mucor*, *Rhizopus*, as well as *Penicillium* and *Aspergillus*.

## 4. Materials and Methods

### 4.1. Essential Oil

*Thymus serpyllum* EO was purchased from Hanus, s.r.o. (Nitra, Slovakia) and was prepared by steam distillation of dried flowering stalk. It was stored in the dark at 4 °C throughout the analyses.

### 4.2. Tested Microorganisms

Microorganisms (*Bacillus subtilis* CCM 2772, *Pseudomonas aeruginosa* CCM 1959, *Yersinia enterocolitica* CCM 5671, *Staphylococcus aureus* subsp. *aureus* CCM 2461, *Enterococcus faecalis* CCM 4224, *Salmonella enteritidis* subsp. *enteritidis* CCM 4420, *Candida krusei* CCM 8271, *Candida albicans* CCM 8186, *Candida tropicalis* CCM 8223, and *Candida glabrata* CCM 8270) were obtained from the Czech collection of microorganisms. The biofilm-forming bacteria *Bacillus subtilis* and *Stenotrophomonas maltophilia* were obtained from the dairy industry and identified with 16S rRNA sequencing and MALDI-TOF MS Biotyper. There were three types of fungi isolated from grape samples, *Penicillium expansum* (from Rheinriesling), *Penicillium crustosum* (from Alibernet), and *Penicillium citrinum* (from Dornfelder). These fungi were identified with 16S rRNA sequencing and MALDI-TOF MS Biotyper.

### 4.3. Chemical Characterization of Essential Oil Samples by Gas Chromatography/Mass Spectrometry (GC/MS) and Gas Chromatography (GC-FID)

GC/MS analysis of selected essential oil samples was performed using an Agilent 6890N gas chromatograph (Agilent Technologies, Santa Clara, CA, USA) coupled to a quadrupole mass spectrometer 5975B (Agilent Technologies, Santa Clara, CA, USA) with an HP-5MS capillary column (30 m × 0.25 mm × 0.25 µm). The temperature program was 60 °C to 150 °C (increasing rate, 3 °C/min) and 150 °C to 280 °C (increasing rate, 5 °C/min). The total run time was 60 min. Helium 5.0 was used as the carrier gas with flow rate of 1 mL/min. The injection volume was 1 µL (the EO sample was diluted in pentane), while the split/splitless injector temperature was set at 280 °C. The investigated sample was injected in the split mode with split ratio at 40.8:1. Electron-impact mass spectrometric data (EI-MS; 70 eV) were acquired in scan mode over the m/z range 35–550. The MS ion source and MS quadrupole temperatures were 230 °C and 150 °C, respectively. Acquisition of data started after a solvent delay time of 3 min. GC-FID analyses were performed on an Agilent 6890N gas chromatograph coupled to a FID detector. The column (HP-5MS) and chromatographic conditions were the same as for GC-MS. The temperature of the FID detector was set at 300 °C.

The individual volatile constituents of injected essential oil samples were identified according to their retention indices [48] and they were compared with the reference spectra (Wiley and NIST databases). The retention indices were experimentally determined by the standard method described in the study by Van Den Dool and Kratz [49], which included retention times of n-alkanes (C6–C34), injected under the same chromatographic conditions. The percentages of the identified compounds (amounts higher than 0.1%) were derived from their GC peak areas.

### 4.4. Antioxidant Activity—DPPH Method

This method is based on colorimetric reaction where 2,2-diphenyl-1-picrylhydrazyl (DPPH, Sigma Aldrich, Germany) changes its color from purple to light yellow due to reaction with antioxidant. Analysis was carried out in a 96-well microplate. The DPPH working solution was prepared from stock solution (0.0025 g/L of DPPH in methanol) and it was diluted ten times. Absorbance was adjusted to 0.7 at wavelength 515 nm. A calibration curve was prepared with standard solution of Trolox (Sigma Aldrich, Schnelldorf, Germany) dissolved in methanol (Uvasol^®^ for spectroscopy, Merck, Darmstadt, Germany) in the concentration range 0–100 µg/mL with 5 μL of essential oil added to 195 μL of DPPH working solution. Blank measurement was prepared by 200 μL of DPPH. The calibration solution was prepared by addition of 5 μL of Trolox solution with corresponding concentration to 195 μL of DPPH working solution [50]. The microplate was incubated for 30 min on a shaker (IKA Inc., Staufen im Breisgau, Germany) at 200 rpm in the dark and studied at 515 nm with a Glomax spectrophotometer (Promega Inc., Madison, WI, USA). The percentage of inhibition was calculated as (A0 − AA)/A0 × 100, where A0 was absorbance of the blank measurement and AA was absorbance of the sample. Antioxidant activity was expressed as antioxidant activity of Trolox related to 1 mL of sample (TEAC).

### 4.5. Antimicrobial Activity—Disc Diffusion Method

Antimicrobial activity of *T. serpyllum* EO was determined using the disc diffusion method. Bacteria were aerobically cultivated on Tryptone Soya Agar (TSA, Oxoid, Basingstoke, UK) at 37 °C for 24 h, yeast at 25 °C for 24 h. An inoculum with an optical density of 0.5 McFarland standard (corresponding to 1.5 × 108 CFU/mL) was prepared and an amount of 100 μL was used for Mueller Hinton agar (MHA, Oxoid, Basingstoke, UK) inoculation. Clean discs with 6 mm diameter were saturated with 10 μL of *T. serpyllum* EO and placed on the agar. Bacteria were incubated aerobically at 37 °C for 24 h and yeast were incubated at 25 °C for 24 h. Criteria for detection of inhibitory activity were: an inhibition zone diameter above 5 mm—weak inhibitory activity, above 10 mm—moderate inhibition, and above 15 mm—very strong inhibition. Each test was repeated three times.

### 4.6. Minimum Inhibitory Concentrations (MIC)

Microorganisms were aerobically cultured for 24 h in Mueller Hinton Broth (MHB, Oxoid, Basingstoke, UK) at 37 °C for bacteria and at 25 °C for yeasts. The 50 µL of microbial suspension with optical density 0.5 McFarland standard was applied to a 96-well microtiter plate. The amount of 100 μL of MHB with *T. serpyllum* EO in concentrations from 400 μL/mL to 0.2 μL/mL, prepared with serial dilution, was added to sample. The contents of the wells were mixed by pipetting. The MHB and EO were used as a negative control, and the MHB with inoculum was used as positive control of the maximal growth.

The MIC of biofilms was measured after 24 h with use of crystal violet. The suspension with non-attached cells was discarded and wells were washed with distilled water three times, dried at room temperature, stained with crystal violet (200 μL 0.1% (*w*/*v*)) for 15 min, and repeatedly washed and dried. Stained biofilms were resolubilized with 200 μL of 33% acetic acid [51]. Absorbance was measured at 570 nm (Glomax spectrophotometer, Promega Inc., Madison, WI, USA). The concentration of EO which had absorbance lower than the absorbance of the maximal growth control was determined as the minimum inhibitory concentration. Each test had three replications.

### 4.7. Analysis of Differences in Biofilm Development with MALDI-TOF MS Biotyper

The various phases of biofilm development were evaluated with MALDI-TOF MS Biotyper. *S. maltophilia* was used as a representative of Gram-negative biofilm-forming bacteria and *B. subtilis* was representative of Gram-positive biofilm-forming bacteria and both bacterial strains were isolated from the milk industry. The main goal was to monitor changes in the structure of the biofilm on glass and wooden surfaces after treatment with EO of *T. serpyllum*. Experimental and control samples were prepared in 50 mL polypropylene tubes with 20 mL of MHB, a wooden toothpick, and a glass slide. The experimental groups contained MHB enriched with 0.5% *T. serpyllum* EO and inoculated samples were incubated at 37 °C on a slope 45° shaker at 170 rpm. Biofilm and planktonic cell samples were analysed on days 3, 5, 7, 9, 12, and 14. The biofilm samples were taken from a glass slide and wooden toothpick using a sterile cotton swab and imprinted onto a MALDI-TOF metal target plate on certain day. The planktonic cells were obtained by removing 300 µL of culture medium. The culture medium was centrifuged for 1 min at 12,000 rpm. The supernatant was discarded, the planktonic cells were washed three times with ultrapure water. The pellet was resuspended in 30 μL of ultrapure water and centrifuged for 1 min at 12,000 rpm. The planktonic cells were resuspended and 1 μL of the suspension was applied to a target plate after washing. The amount of 1 μL of α-Cyano-4-hydroxycinnamic acid matrix (10 mg/mL) was applied to biofilm and planktonic cell samples and dried at room temperature. The samples were processed with MALDI-TOF MicroFlex (Bruker Daltonics) linear and positive mode for the range of m/z 200–2000 after crystallization. The spectra were obtained by an automatic analysis and the same sample similarities were used to generate the standard global spectrum (MSP). Nineteen MSP from the spectra generated by the MALDI Biotyper 3.0 were grouped into dendrograms using Euclidean distance [35].

### 4.8. Antimicrobial Analysis of Bread In Situ

The in situ antimicrobial effects of *T. serpyllum* EO against biofilm-forming bacteria (*S. maltophilia and B. subtilis*) and fungi (*Penicillium* spp.) were analyzed on bread in order to inhibit food degradation by pathogens. The bread was sliced (15 × 15 × 1.5 cm) and placed in 0.5 L sterile glass jars (Bormioli Rocco, Parma, Italy). Biofilm-forming bacteria were cultured for 24 h at 37 °C on Tryptone Soya agar (TSA, Oxoid, Basingstoke, UK) and *Penicillium* on Sabouraud Dextrose agar (SDA, Oxoid, Basingstoke, UK) at 25 °C for 5 days before inoculation. The inoculum was applied to the bread by three stabs. A sterile filter paper (6 cm in diameter) was placed under the jar lid and 100 µl of *T. serpyllum* EO (62.5, 125, 250, and 500 µL/L + ethyl acetate) were applied. The control group of bread was without EO treatment. The jars were hermetically sealed and maintained at 25 °C ± 1 °C for 14 days in the dark.

In situ bacterial growth was determined using stereological methods. In this concept, the volume density (Vv) of bacterial colonies was firstly estimated using ImageJ software counting the points of the stereological grid hitting the colonies (P) and those (p) falling to the reference space (growth substrate used). The volume density of bacterial colonies was consequently calculated as follows: Vv (%) = P/p. The antibacterial activity of EO was defined as the percentage of bacterial growth inhibition (BGI) BGI = [(C − T)/C] × 100, where C and T were bacterial growth (expressed as Vv) in the control group and the treatment group, respectively. The negative results represented the growth stimulation.

The size of microfungal colonies with visible growth and sporulation was assessed with stereological methods. In this concept, the volume density of the colonies was firstly assessed using ImageJ software counting the points of the stereological grid hitting the colonies and those falling to the reference space (growth substrate used). EO antifungal activity was expressed as mycelial growth inhibition in % (MGI): MGI = [(C − T)/C] × 100, where C was fungal growth (expressed as volume density) in the control group and T was the growth in the treatment group [52,53].

### 4.9. In Situ Antimicrobial Analysis on Carrots

In situ antimicrobial analysis in the vapor phase was tested on biofilm-forming bacteria *S. maltophilia* and *B. subtilis* and fungi *Penicillium* spp. Warm MHA was poured into 60 mm petri dishes (PD) and the lid. Sliced carrots (0.5 mm) were placed on agar. Then, an inoculum was prepared as previously described. *T. serpyllum* EO was diluted twice in ethyl acetate to 500, 250, 125, and 62.5 μL/L and used for sterile filter paper inoculation. The filter paper was placed in for 1 min to evaporate the remaining ethyl acetate, sealed and incubated for at 37 °C for 7 days and for fungi at 25 °C for 14 days. Growth assessment was performed as in the in situ antimicrobial activity method.

### 4.10. Statistical Data Evaluation

SAS^®^ software used for data processing. The results of the MIC value (concentration that caused 50% and 90% inhibition in bacterial growth) were determined by logit analysis.

## 5. Conclusions

The MALDI-TOF MS Biotyper used for the first time in analyses of a biofilm formed by *S. maltophilia* and *B. subtilis* as contaminant in the food industry in our study. The inhibition of the biofilm after *T. serpyllum* application was also observed for the first time. To date, only a limited number of studies on the biological activity of *T. serpyllum* essential oil have been published. The main components of the EO of *T. serpyllum* were thymol, 18.8%; carvacrol, 17.4%; o-cymene, 15.4%; and geraniol 10.7%. The antioxidant activity of EO was high at 82.4 ± 0.5%, which corresponds to 463.53 ± 2.60 µg Trolox/mL sample. *T. serpyllum* EO had very good antimicrobial effects as well as antibiofilm effects observed on various surfaces and detected by MALDI-TOF MS Biotyper. The MALDI-TOF MS Biotyper was a suitable method for evaluating phases of biofilm development. *T. serpyllum* EO demonstrated inhibitory activity on microorganisms in a food model in the vapor phase. The present study showed some new findings on the characteristics of *T. serpyllum* EO which should be a suitable alternative to extend the shelf-life of root vegetables and to the chemical control of *Penicillium* on bread.

## Figures and Tables

**Figure 1 plants-10-01416-f001:**
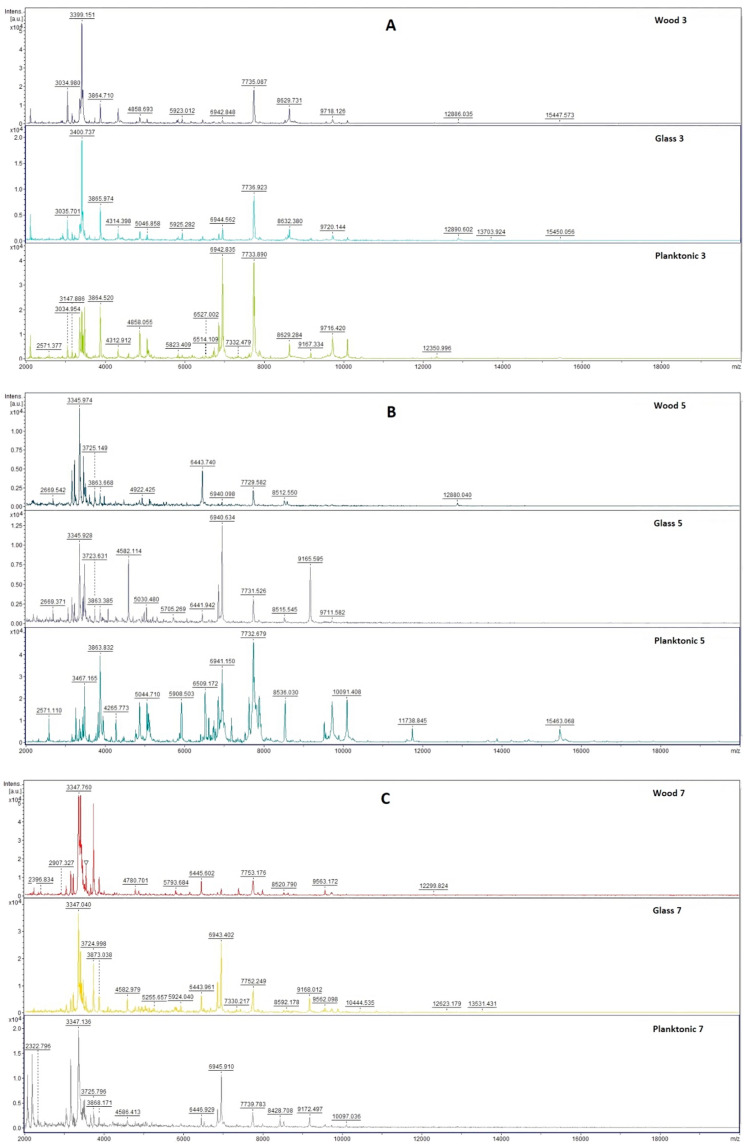
MALDI-TOF mass spectra of *S. maltophilia* biofilm during development: (**A**) 3rd day, (**B**) 5th day, (**C**) 7th day, (**D**) 9th day, (**E**) 12th day, and (**F**) 14th day.

**Figure 2 plants-10-01416-f002:**
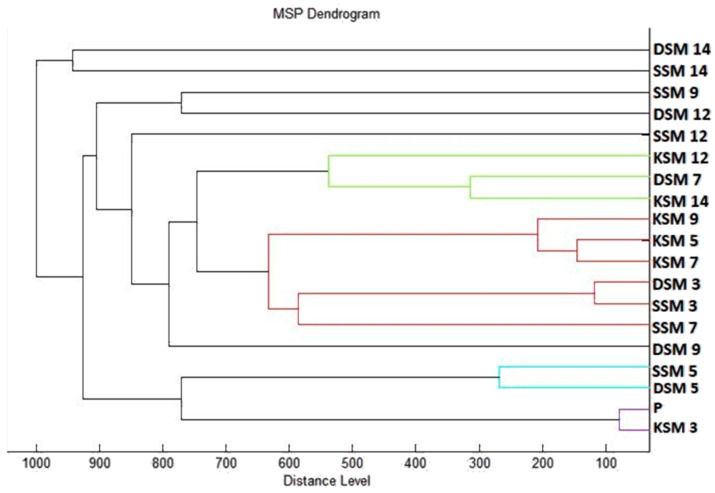
Dendrogram of *S. maltophilia* generated using MSPs of the planktonic cells and the control. **SM,** S. maltophilia; **K,** control; **S,** glass; **D,** wood; and **P,** planktonic cells.

**Figure 3 plants-10-01416-f003:**
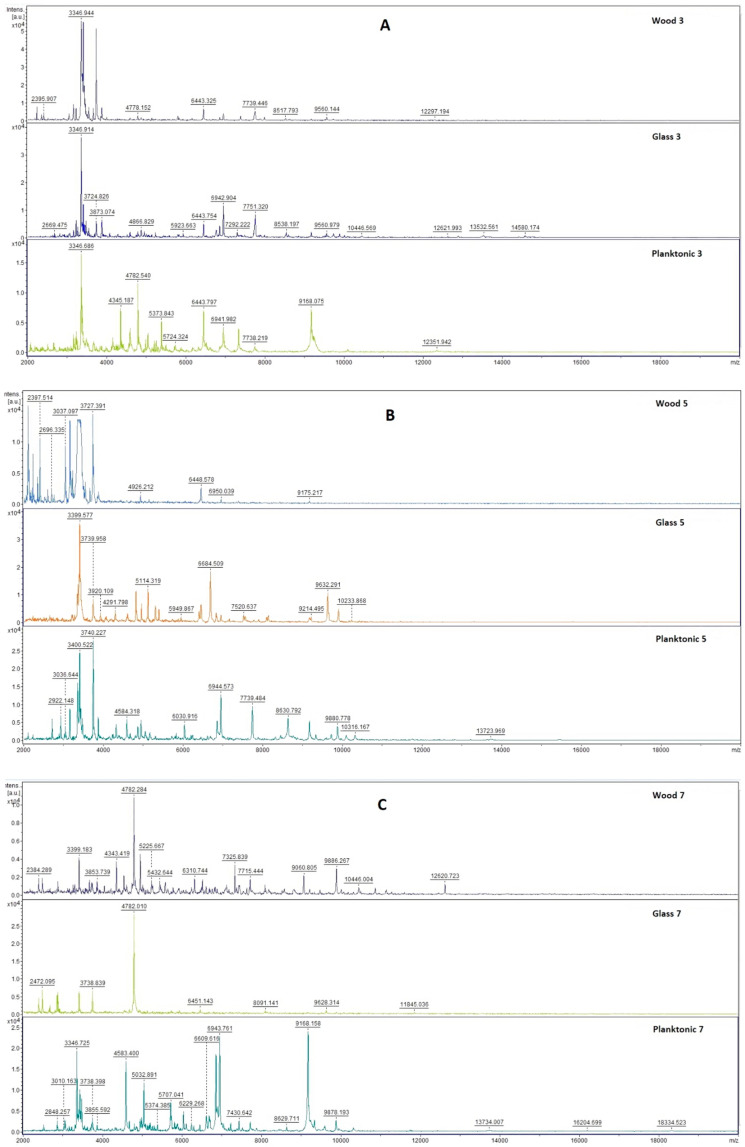
Representative MALDI-TOF mass spectra of *B. subtilis*: (**A**) 3rd day, (**B**) 5th day, (**C**) 7th day, (**D**) 9th day, (**E**) 12th day, and; (**F**) 14th day.

**Figure 4 plants-10-01416-f004:**
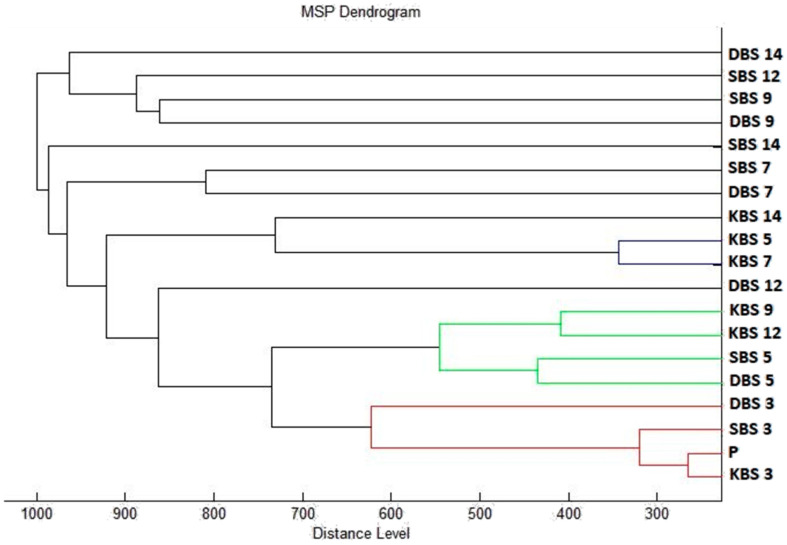
Dendrogram of *B. subtilis* generated using MSPs for the planktonic cells and the control. **BS,**
*B. subtilis*; **K,** control; **S,** glass; **D,** wood; **P,** planktonic cells.

**Table 1 plants-10-01416-t001:** Chemical composition of essential oil of *Thymus serpyllum*.

Number	RI ^a^	Compound ^b^	% ^c^
1	858	cis-hexen-1-ol	Tr
2	926	α-thujene	0.5
3	938	α-pinene	1.2
4	948	camphene	1.1
5	976	1-octen-3-one	0.8
6	980	β-pinene	0.3
7	992	β-myrcene	1.1
8	993	octan-3-ol	Tr
9	1004	α-phellandrene	Tr
10	1009	δ-3-carene	Tr
11	1016	α-terpinene	1.1
12	1026	*o*-cymene	15.4
13	1028	α-limonene	1.2
14	1033	1,8-cineole	1.5
15	1060	γ-terpinene	8.1
16	1088	α-terpinolene	Tr
17	1098	Linalool	5.3
18	1148	Camphor	0.9
19	1170	Borneol	2.3
20	1178	4-terpineol	1.8
21	1235	Thymol methyl ether	Tr
22	1241	Carvone	Tr
26	1245	Carvacrol methyl ether	0.5
27	1255	Linalool acetate	1.5
28	1256	Geraniol	10.7
29	1286	Bornyl acetate	0.6
30	1290	Thymol	18.8
31	1302	Carvacrol	17.4
32	1353	α-cubebene	Tr
33	1360	Eugenol	Tr
34	1379	α-copaene	Tr
35	1380	Geranyl acetate	4.4
36	1406	Methyl eugenol	Tr
37	1422	cis-caryophyllene	2.4
38	1456	α-humulene	1.0
39	1443	Aromadendrene	Tr
40	1507	β-bisabolene	Tr
41	1525	δ-cadinene	Tr
42	1583	Caryophyllene oxide	Tr
	Total		99.9

^a^ Values of retention indices on HP-5MS column; ^b^ identified compounds; ^c^ tr-compounds identified in amounts less than 0.1%.

**Table 2 plants-10-01416-t002:** Antimicrobial activity of essential oil of *T. serpyllum*.

Microorganism	Inhibition Zone (mm)	Activity of EO	MIC 50 (µL/mL)	MIC 90 (µL/mL)	Activity of EO
*Salmonella enteritidis*	15.67 ± 1.53	***	0.39	0.78	***
*Pseudomonas aeruginosa*	30.33 ± 0.58	***	0.20	0.39	***
*Yersinia enterocolitica*	6.33 ± 0.58	*	12.5	25.00	*
*Staphylococcus aureus*	8.33 ± 1.15	*	12.5	25.00	*
*Bacillus subtilis*	11.33 ± 1.53	**	6.25	12.50	**
*Enterococcus faecalis*	13.67 ± 1.53	**	6.25	12.50	**
*Candida albicans*	12.33 ± 1.53	**	1.56	3.13	***
*Candida krusei*	11.00 ± 1.00	**	3.13	6.25	***
*Candida tropicalis*	9.33 ± 0.58	*	12.5	25.00	*
*Candida glabrata*	11.00 ± 1.00	**	3.13	6.25	***
Biofilm *Stenotrophomonas maltophilia*	15.67 ± 0.58	***	0.39	0.78	***
Biofilm *Bacillus subtilis*	25.33 ± 0.58	***	0.20	0.39	***

* Weak antimicrobial activity, ** moderate inhibitory activity, *** very strong inhibitory activity.

**Table 3 plants-10-01416-t003:** In situ analysis of the antibacterial activity of the vapor phase of *T. serpyllum* essential oil in bread.

Bacterial Growth Inhibition (%)				
Concentration of EO	62.5 µL/L	125 µL/L	250 µL/L	500 µL/L
Microorganisms				
*S. maltophilia*	11.13 ± 0.88	−55.51 ± 2.07	44.05 ± 1.08	83.51 ± 0.54
*B. subtilis*	−12.38 ± 1.22	30.50 ± 1.76	94.47 ± 2.73	84.21 ± 1.83
**Mycelial Growth Inhibition (%)**				
**Concentration of EO**	**62.5 µL/L**	**125 µL/L**	**250 µL/L**	**500 µL/L**
**Microorganisms**				
*P. citrinum*	84.93 ± 1.11	58.93 ± 1.97	73.27 ± 2.07	73.51 ± 1.33
*P. crustosum*	15.21 ± 0.43	42.63 ± 1.54	48.85 ± 2.63	54.15 ± 1.46
*P. expansum*	34.34 ± 2.41	84.93 ± 0.55	77.65 ± 1.89	66.26 ± 1.42

**Table 4 plants-10-01416-t004:** Results of in situ analysis of antibacterial activity of the vapor phase of *T. serpyllum* essential oil on carrots.

Bacterial Growth Inhibition (%)				
Concentration of EO	62.5 µL/L	125 µL/L	250 µL/L	500 µL/L
Microorganisms				
*S. maltophilia*	3.39 ± 0.09	25.72 ± 1.78	6.07 ± 1.32	83.77 ± 0.66
*B. subtilis*	42.92 ± 1.11	41.67 ± 2.09	47.15 ± 0.24	72.46 ± 2.07
**Mycelial Growth Inhibition (%)**				
**Concentration of EO**	**62.5 µL/L**	**125 µL/L**	**250 µL/L**	**500 µL/L**
**Microorganisms**				
*P. citrinum*	−7.50 ± 0.88	63.27 ± 0.27	9.25 ± 1.75	−7.50 ± 0.73
*P. crustosum*	44.20 ± 1.33	53.82 ± 0.86	23.24 ± 1.92	25.80 ± 1.01
*P. expansum*	0.85 ± 0.09	39.03 ± 1.45	8.75 ± 0.49	16.90 ± 0.61

## Data Availability

Data is contained within the article.

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
