# Peer review of "Thymus serpyllum Essential Oil and Its Biological Activity as a Modern Food Preserver"

_plants, 2021, doi:10.3390/plants10071416_

Round 1
Reviewer 1 Report
Due to their structure, biofilms are a difficult research topic. The essential oil used in the research was commercial, described in the product specification, hence, in my opinion, in Table 1, RI can be omitted, and if so, also RI lit. Usually, in the study of oils, the traces are less than 0.05%. Table 1 gives incorrect systematic nomenclature, for example, it is 3-octanol and it should be octan-3-ol, the note concerns several compounds. The description of the configuration as cis / trans is recommended for cyclic compounds. Thank you for your cooperation.
Author Response
Reviewer #1
Due to their structure, biofilms are a difficult research topic. The essential oil used in the research was commercial, described in the product specification, hence, in my opinion, in Table 1, RI can be omitted, and if so, also RI lit. Usually, in the study of oils, the traces are less than 0.05%. Table 1 gives incorrect systematic nomenclature, for example, it is 3-octanol and it should be octan-3-ol, the note concerns several compounds. The description of the configuration as cis / trans is recommended for cyclic compounds. Thank you for your cooperation.
Response: As we pointed out in the experimental part (The individual volatile constituents of injected essential oil samples were identified according to their retention indices and they were compared with the reference spectra), we did not exclude exp RI values. Reason is simple: true identification of volatile constituents required mass spectra and RI values (in many cases mass spectra of different compounds can be quite similar, so RI values gives valuable informations for identification.
For trace compounds. Yes, we agree that many researchers give trace compounds in conc less than 0.05 %. But, on our opinion (also in many literature data trace compounds were less than 1 %), slightly is precise when authors defined trace compounds for percentage concentrations less than 1 % (it is also technical advice by Agilent representatives).
In accordance with reviewer comment, we changed names of few compounds in Table 1:
cis-3-hexenol to cis-hexen-1-ol,
3-octanol to octan-3-ol,
(E)-caryophyllene to cis-caryophyllene.
Reviewer 2 Report
1. wonder whether in the introduction the authors could add some more references on the already knowledge on antibiofilm activity of essential oil of Thymus serpyllum.
2. At the end of the introduction is not clear which are the innovative aspects of the article since we cannot consider that chemical composition, antioxidant, and antimicrobial of essential oil of Thymus serpyllum is a novel topic.
3. This part 4.9. Statistical Data Evaluation: does not contain all the required information
better to contain more details about the used statistical methods like (Dendrogram) and the programs used in the data analysis.
Author Response
Reviewer #2
Point 1: wonder whether in the introduction the authors could add some more references on the already knowledge on antibiofilm activity of essential oil of Thymus serpyllum.
Response: References about antibiofilm activity of T. serpyllum EO were added (line 47-48).
Point 2: At the end of the introduction is not clear which are the innovative aspects of the article since we cannot consider that chemical composition, antioxidant, and antimicrobial of essential oil of Thymus serpyllum is a novel topic.
Response: The aim of the research was enhanced. (line 68-72) This is the first time, when EO from Slovak republic was tested. We tried to highlight the use of MALDI for biofilm structure changes.
Point 3: This part 4.9. Statistical Data Evaluation: does not contain all the required information better to contain more details about the used statistical methods like (Dendrogram) and the programs used in the data analysis.
Response: We did not use the statistical analysis during dendrogram construction. The program that was used is mentioned in section 4.7.
Reviewer 3 Report
This study tests chemical composition and antimicrobial activity of essential oil from Thymus serpyllum. The introduction provides some information about properties of thyme essential oil, but should be a little developed. What type of microorganism is sensitive to thyme essential oil?
The objective of the study is clearly defined. The experimental apparatus is appropriate for the study. Methods are well described and provide sufficient information to reproduce the experiments. The results are clearly explained and presented in an appropriate format. Conclusions result from the conducted research.
Minor remarks:
The authors should supplement the manuscript with the variety of thyme.
The authors should elaborate on the description of the distillation method and provide the efficiency of the process.
Author Response
Reviewer #3 This study tests chemical composition and antimicrobial activity of essential oil from Thymus serpyllum. The introduction provides some information about properties of thyme essential oil, but should be a little developed. What type of microorganism is sensitive to thyme essential oil?
Response: The types of microorganisms sensitive to thyme EO were added (line 47).
The objective of the study is clearly defined. The experimental apparatus is appropriate for the study. Methods are well described and provide sufficient information to reproduce the experiments. The results are clearly explained and presented in an appropriate format. Conclusions result from the conducted research.
Point 1: The authors should supplement the manuscript with the variety of thyme.
Response: The manufacturer did not provide the information about thyme variety.
Point 2: The authors should elaborate on the description of the distillation method and provide the efficiency of the process.
Response: The T. serpyllum EO was bought from commercial company, and they did not provide the information about distillation method or yield. We asked the manufacturer about the distillation method, bud we are not sure if there is enough time to obtain these information.